# A systematic review of psychosocial functioning changes after gender-affirming hormone therapy among transgender people

David Matthew Doyle ●[1] ✉, Tom O. G. Lewis ●[2] & Manuela Barreto ●[2]

This systematic review assessed the state and quality of evidence for effects of gender-affirming hormone therapy on psychosocial functioning. Forty-six relevant journal articles (six qualitative, 21 cross-sectional, 19 prospective cohort) were identified. Gender-affirming hormone therapy was consistently found to reduce depressive symptoms and psychological distress. Evidence for quality of life was inconsistent, with some trends suggesting improvements. There was some evidence of affective changes differing for those on masculinizing versus feminizing hormone therapy. Results for self-mastery effects were ambiguous, with some studies suggesting greater anger expression, particularly among those on masculinizing hormone therapy, but no increase in anger intensity. There were some trends toward positive change in interpersonal functioning. Overall, risk of bias was highly variable between studies. Small samples and lack of adjustment for key confounders limited causal inferences. More high-quality evidence for psychosocial effects of gender-affirming hormone therapy is vital for ensuring health equity for transgender people.

The most common form of medical intervention sought by transgender people (here we use this term to refer to anyone whose gender identity does not match their gender assigned at birth, for example, including non-binary, gender fluid and genderqueer people) is gender-affirming hormone therapy[1,2]. For example, approximately 60%–70% of those who attended the most widely used gender identity clinic in the Netherlands between 2010 and 2014 began hormone therapy within 5 years (ref. 3). Once on gender-affirming hormone therapy, transgender people are generally instructed to continue to use some dosage of gender-affirming hormones throughout their lives[4,5]. Transgender people tend to engage with these therapies to modify their physical presentation in line with their gender identity[6,7]; importantly, sex hormones may also affect psychological states and social interactions, as is observed during puberty[8]. However, despite

the large and growing prevalence of gender-affirming hormone therapy across countries[9], no systematic review of research has been conducted to examine the state and quality of evidence for effects of gender-affirming hormone therapy on psychosocial functioning among transgender people. Research in humans and non-human animals has suggested that hormones may influence psychosocial functioning via biological pathways[10–13], but findings in this literature have often been mixed or inconclusive. There is a pressing need to better understand the psychosocial consequences of hormones, particularly given the critical implications for transgender health.

Psychosocial functioning is a core facet of human life that shapes how people relate to others and the quality of their social relationships. Psychosocial functioning refers to a variety of traits, characteristics and dispositions that have been broadly classified[14] as

[1]Department of Medical Psychology, Amsterdam University Medical Centers, Location VUmc, Amsterdam, the Netherlands. [2]Department of Psychology, University of Exeter, Exeter, UK. ✉e-mail: d.m.doyle@amsterdamumc.nl

(1) well-being (for example, self-acceptance, positive mood, satisfaction with life), (2) self-mastery (for example, self-control, low aggression and impulsivity) and (3) interpersonal functioning (for example, trust, secure attachment, empathy). Better psychosocial functioning across these three domains has been shown to be associated with healthier experiences in social relationships, including higher quality romantic relationships[15] and friendships[16], potentially decreasing social isolation and loneliness throughout the life course[17]. Worryingly, accumulating evidence points toward impairments in psychosocial functioning and greater negative experiences in social relationships among transgender people relative to cisgender populations, likely driven in part by experiences of stigma and invalidation[18]. For example, on average, transgender people report higher levels of social anxiety relative to cisgender people[19]. Decades of research have shown that social relationships are vital to health and well-being[20,21], including longevity and mortality[22]. In perhaps the starkest example of how poor psychosocial functioning and negative experiences in social relationships may affect transgender people, suicide rates are substantially elevated in this group relative to cisgender people. Approximately one in three transgender people attempt suicide in their lifetime[23], and past work has linked this risk in part to disruption in their social life[24,25].

Transgender people can seek hormone therapies to affirm their gender identity, to aid with gender congruence and alleviate gender dysphoria. Gender-affirming hormone therapy usually involves exogenous administration of testosterone for transmasculine people, and oestradiol along with an anti-androgen for transfeminine people, as well as individualized treatments for those identifying as non-binary or gender diverse. Evidence suggests that gender-affirming hormone treatments reorganize brain structure and function[26], which is in line with evidence demonstrating the general importance of sex steroids to neurobiology[27,28]. These fundamental neurobiological changes may in turn shape responses to social stimuli and general social functioning via biological processes[29].

A growing body of research, principally from the field of social neuroendocrinology, has suggested that both endogenous and exogenous hormones influence psychosocial functioning via biological pathways[10,12,13]. A relatively large number of studies have found that exogenous administration of testosterone may affect various aspects of psychosocial functioning in cisgender men and women, including potentially increasing social aggression and decreasing emotion identification and trust, although such findings are still often tentative and inconclusive[30]. Comparatively less research has been devoted to studying the psychosocial effects of exogenous administration of oestrogens in humans, but some work suggests that they may improve mood in cisgender women, particularly for those diagnosed with depressive disorders[31]. Additionally, exogenous administration of progesterone has been linked to changes in mood in cisgender women in both positive and negative ways, depending on dose and other factors, such as history of premenstrual syndrome[32].

Clinical guidance for the provision of gender-affirming care, particularly gender-affirming hormone therapy, relies on a relatively underdeveloped and somewhat inconsistent evidence base with little acknowledgement of potential psychosocial implications[33,34]. At best, in addition to potential medical side-effects (for example, increased risk of deep vein thrombosis for those on feminizing hormone therapies and increased risk of polycythaemia for those on masculinizing hormone therapies), clinical guidance[1,2] may mention the possibility of some changes in mood or personality, such as increased aggression following commencement of testosterone[35,36]. A few reviews have been conducted on the effects of gender-affirming hormone therapy on mental health and quality of life[37–40], but each of these focused on just one dimension of psychosocial functioning (that is, well-being). Speaking to the societal relevance of this topic, some critics have pointed to a perceived lack of evidence for various outcomes in practice

## Table 1 | Adjustment for potential confounding by study

| Confounder | k (%) |
|---|---|
| Body image | 12 (30) |
| Gender-affirming surgeries | 9 (23) |
| Sexuality | 6 (15) |
| Suicidal ideation | 5 (13) |
| Non-binary participants included | 4 (10) |
| Social support | 3 (8) |
| Alcohol and substance use | 2 (5) |
| Autism | 2 (5) |
| Cognitive functioning and reasoning (for example, visual and verbal tasks) | 2 (5) |
| Gender affirmation from others | 2 (5) |
| Personal views on hormone therapy | 2 (5) |
| Puberty blockers | 2 (5) |
| Self-harm | 2 (5) |
| Social stigma | 2 (5) |
| Assertiveness | 1 (3) |
| Body mass index | 1 (3) |
| Communication style | 1 (3) |
| Eating disorders | 1 (3) |
| Gender identity disclosure | 1 (3) |
| Honest responding | 1 (3) |
| Negative life events | 1 (3) |
| Paranoid ideation | 1 (3) |
| Psychiatric hospitalization | 1 (3) |
| Psychopharmacological treatments (for example, anti-depressants, anti-anxiety medication, etc.) | 1 (3) |
| Relationship status | 1 (3) |
| Ruminating disorders | 1 (3) |
| Social transition | 1 (3) |
| Subjective dysphoria | 1 (3) |

guidelines for gender-affirming hormone therapies as a reason to limit their use[41,42]. Therefore, a systematic review of changes in psychosocial functioning is useful to highlight outcomes for which there is strong evidence as well as to instigate further research on outcomes that need clarification.

The aim of the current review was to evaluate the state and quality of evidence for effects of gender-affirming hormone therapy on a wider range of dimensions of psychosocial functioning among transgender people. We sought to include studies with diverse research methodologies. The primary aim was to examine the strength of evidence for potential causal effects on psychosocial functioning resulting from gender-affirming hormone therapy. That is, we aimed to understand not only what effects hormone therapy might have on psychosocial functioning, but also whether these could be unambiguously and directly related to the hormonal changes, or whether existing evidence does not separate this from the various other changes that are associated with gender transition. To do so, we also formally assessed risk of bias for each study included in our review using the Newcastle-Ottawa Scale[43]—see Supplementary Tables 1 and 2 for full coding of each quantitative study—as well as coding for potential confounding variables (see Table 1) to gauge which were most often included in the quantitative literature on this topic.

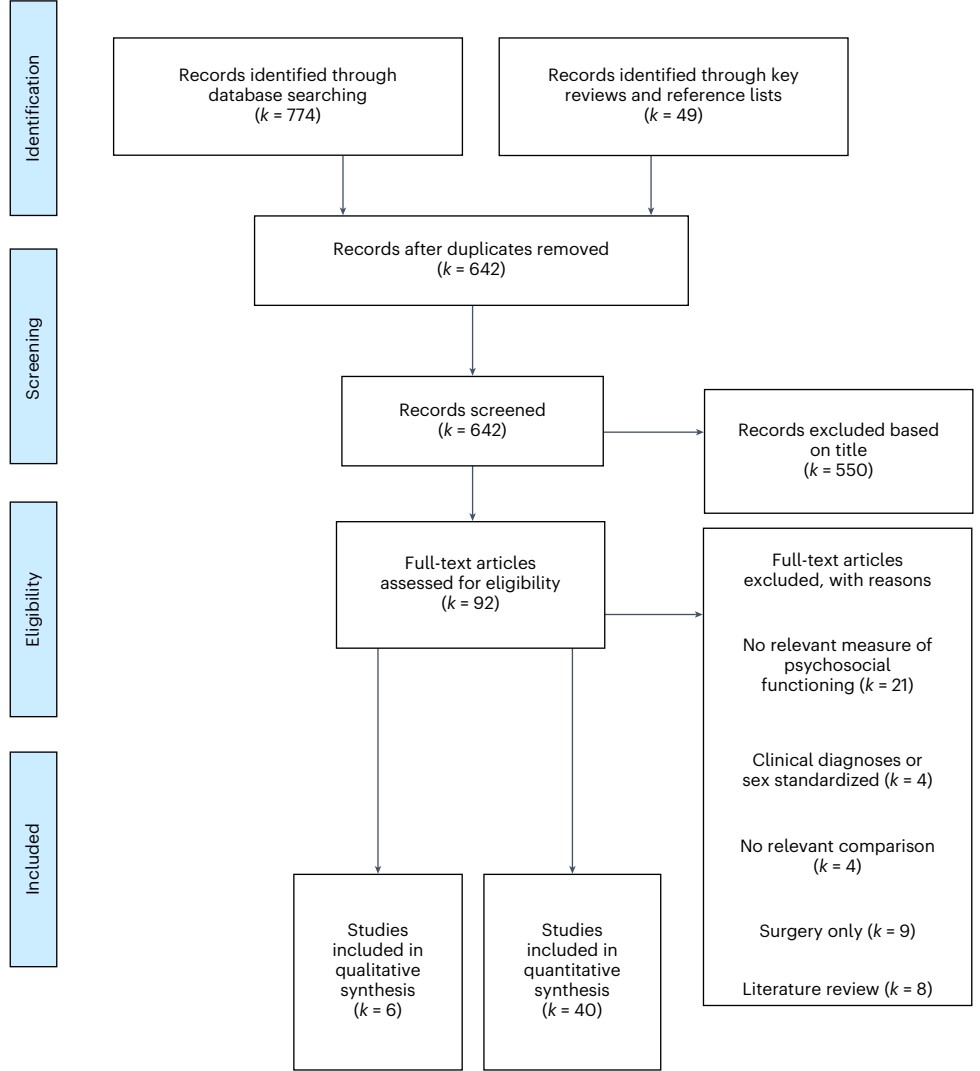

**Fig. 1 | Study selection for systematic review.** PRISMA study inclusion flowchart.

## Results

### Qualitative review

We identified only six (13% of all studies) qualitative studies (see Fig. 1 for study inclusion flowchart), all involving semistructured interviews. The total number of transgender participants across the qualitative studies was 171, with individual sample sizes ranging from 10 to 67 transgender participants per study. The qualitative research identified (*k* = 6, where *k* is the number of studies) predominantly involved participants on feminizing hormone therapy[44–49], with only two studies[44,47] including participants on masculinizing hormone therapy. In the following sections we briefly summarize results related to changes in psychosocial functioning from the qualitative literature, organized by the experiences of those on feminizing and masculinizing hormone therapy separately.

**Experiences on feminizing hormone therapy.** Overall, the qualitative literature tended to support positive changes in well-being among people after starting feminizing hormone therapy, although often with the qualification that improvements in well-being were attributed to satisfaction with changes in appearance rather than to direct effects of hormones on psychosocial states[45,46,48]. More specifically, in addition to reductions in distress[48] and depressive symptoms[49], participants reported improvements in self-image and self-acceptance[44,46,47] and

less self-monitoring[48] after beginning gender-affirming hormone therapy. There were also indications of changes in emotional functioning, generally experienced as positive and related to greater emotional range and freedom-of-expression[46,48], but sometimes noted as negative and related to mood swings and emotional imbalance[45,46,49]. Finally, improvements in interpersonal functioning and the quality of relationships were commonly reported after beginning gender-affirming hormone therapy[44–46], but again these were also largely attributed to changes in satisfaction with appearance rather than direct effects of hormones[45,46]. While a desire for changes to appearance is a core aspect of seeking gender-affirming hormone therapy for many transgender people[7], understanding this as a separate pathway (that is, a psychological one) can help to isolate any potential biological pathway through which hormones might affect psychosocial functioning.

**Experiences on masculinizing hormone therapy.** Qualitative evidence for changes in psychosocial functioning among people on masculinizing hormone therapy came from only two studies[44,47]. In one study[44], participants described increased confidence and assertiveness, but also concern over increased difficulty controlling anger and reduced emotional openness. A second study[47] included participants on masculinizing hormone therapy in the overall sample and, although no quotes from these participants were specifically highlighted in the

subtheme related to gender-affirming hormone therapy and sense of self, overall participants reported improvements in self-image and self-acceptance.

### Quantitative review

**Cross-sectional studies.** Twenty-one cross-sectional studies (46% of all studies included in this review) met the inclusion criteria for our review (see Fig. 1 for study inclusion flowchart). The total number of transgender participants across the cross-sectional studies was 37,913, with individual sample sizes ranging from 42 to 21,598 transgender participants per study. The most common method of recruitment in the cross-sectional studies was general online convenience sampling ($k = 9$; 43%), while four studies (19%) used local convenience sampling (for example, from cities and universities) and eight (38%) used hospital/clinic-based sampling. Most studies ($k = 16$; 76%) compared transgender people who were on, or had undergone, gender-affirming hormone therapy with those who had never done so, but four studies (19%) compared transgender people on gender-affirming hormone therapy with either those planning to undergo gender-affirming hormone therapy or those currently waitlisted. While 20 (95%) of these studies included assessments of well-being (the majority of studies focused on depressive symptoms), only one (5%) included a measure relevant to self-mastery and seven (33%) included measures relevant to interpersonal functioning.

**Feminizing hormone therapy.** *Well-being.* For those on feminizing hormone therapy, seven cross-sectional studies[50–56] demonstrated lower levels of depressive symptoms relative to controls, while one study[57] with moderate risk of bias, with a sample composed of 178 transgender women who were United States Armed Forces veterans, and one study[58] with low risk of bias, with 71 transgender women over the age of 50 recruited from a national gender identity clinic in the United Kingdom, found no significant differences—although neither of these studies included a power analysis. Additionally, one study[59] with moderate risk of bias using the 2015 US Transgender Survey, with 8,827 transgender women and 1,104 genderqueer and non-binary people assigned male at birth, showed lower levels of psychological distress for those on feminizing hormone therapy relative to controls.

While three studies[51,52,58] showed lower levels of anxiety in those on feminizing hormone therapy relative to controls, three other studies[50,55,56] showed no significant differences, although each of these studies, with moderate risk of bias, utilized different measures of anxiety and none included a power analysis.

Four studies[53,60–62] found better quality of life for those on feminizing hormone therapy, while four other studies[63–66] failed to confirm this effect, with three of these studies using the same outcome measure, the Short Form Health Survey (SF-36). Note, however, that two of these studies[63,64], with moderate risk of bias, and conducted in Thailand by the same group, included just 44 and 60 transgender women, respectively, and no power analysis, with participants also using hormones that had not been prescribed to them or supplied by a medical professional.

The same two studies from Thailand[63,64], with moderate risk of bias, reported no significant differences in optimism. Self-esteem was found to be higher for those on feminizing hormone therapy relative to controls in two studies[58,61] with low risk of bias.

*Self-mastery.* One study[51] with 208 transgender women in the United States suggested lower levels of anger for those on feminizing hormone therapy relative to controls. A moderate risk of bias was found for this study.

*Interpersonal functioning.* Two studies[52,67] reported lower levels of social anxiety for those on feminizing hormone therapy relative to controls. Another study[56], with moderate risk of bias, found no difference in social anxiety using the same measure[67], the three-item Social

Phobia Inventory (Mini-SPIN), in an online convenience sample with 363 transfeminine people in the United States. Again, the two studies from Thailand[63,64], with moderate risk of bias, suggested no significant differences in terms of social functioning. One study[58] with low risk of bias found fewer interpersonal problems among those on feminizing hormone therapy relative to controls.

*Cross-sectional summary.* To summarize, the most consistent evidence was found for lower levels of depressive symptoms and distress for those on femininizing hormone therapy relative to controls. While less general and social anxiety and greater quality of life are also possible, evidence is less consistent for these outcomes. This may be due to differences in measures used to assess anxiety, the source from which participants in a given study were acquiring hormones (medical professional versus unlicenced provider) and the small samples in some studies, limiting statistical power. Furthermore, evidence for outcomes such as optimism, self-esteem, anger, social functioning and interpersonal problems is limited by the fact that only few studies have examined each of these traits in transgender people undergoing feminizing hormone therapy.

**Masculinizing hormone therapy.** *Well-being.* For people on masculinizing hormone therapy, seven cross-sectional studies[52,54–56,61,68,69] demonstrated lower levels of depressive symptoms relative to controls. One study[57], with moderate risk of bias, and a sample composed of 28 transgender women who were United States Armed Forces veterans, found no significant differences using the Patient Health Questionnaire-9 (PHQ-9). Additionally, one study[59], with moderate risk of bias, using the 2015 US Transgender Survey with 7,595 transgender men and 3,711 genderqueer and non-binary people assigned female at birth, showed lower levels of psychological distress for those on masculinizing hormone therapy relative to controls.

Relatively consistently, four studies[52,56,68,69] demonstrated lower levels of anxiety, but one study[55], with moderate risk of bias, with an online convenience sample in the United States with 234 transmasculine people and 49 non-binary people assigned female at birth, found no significant difference using the Depression, Anxiety and Stress Scale (DASS-21).

Results for quality of life were very mixed, with four studies[53,60,61,70] suggesting higher scores for those on masculinizing hormone therapy relative to controls, two studies[65,66] with moderate and low risk of bias finding no significant differences, and one study[64] from Thailand with moderate risk of bias showing evidence for lower quality of life—albeit in a convenience sample of 60 transgender men who were primarily using hormones that had not been prescribed to them or supplied by a medical professional.

The same study[64] from Thailand reported no significant difference in optimism. Another study[61], with low risk of bias, demonstrated higher self-esteem for those on masculinizing hormone therapy relative to controls in a sample of 31 transmasculine people in France.

*Self-mastery.* No cross-sectional studies focusing on masculinizing hormone therapy included measures relevant to self-mastery.

*Interpersonal functioning.* Consistently, four studies[52,56,67,68] demonstrated lower levels of social anxiety for those on masculinizing hormone therapy relative to controls. One other study[64] suggested no significant difference in terms of social functioning. It should be noted again that this study had a moderate risk of bias and a sample of 60 transgender men primarily using hormones that had not been prescribed to them or supplied by a medical professional.

*Cross-sectional summary.* Taken together, the most consistent evidence is for lower levels of depressive symptoms and distress as well as general and social anxiety for those on masculinizing hormone therapy relative

to controls, indicators of improvements in well-being and interpersonal functioning. Results for quality of life were mixed, with one study even demonstrating lower levels for those on masculinizing hormone therapy relative to controls—one of the few cases of lesser well-being for those on gender-affirming hormone therapy—although this may have been related to the fact that participants in this study were primarily acquiring hormones from unlicenced providers without appropriate medical guidance and supervision. While there were no cross-sectional studies assessing constructs related to self-mastery, evidence for outcomes such as optimism, self-esteem and social functioning is limited by the fact that only a single study has examined each of these traits in transgender people undergoing masculinizing hormone therapy.

**Prospective cohort studies.** Nineteen prospective cohort studies (41% of all studies included in this review) met the inclusion criteria for our review (see Fig. 1 for study inclusion flowchart). The total number of participants across these studies was 3,491. Sample sizes in the studies ranged from 14 to 898 participants per study. The cohorts were recruited from gender identity clinics or medical centres in Italy (k = 5; 26%), the Netherlands (k = 5; 26%), Belgium (k = 2; 11%), the United Kingdom (k = 2; 11%), the United States (k = 1; 5%) and Turkey (k = 1; 5%). Two papers (11%) used the ENIGI cohort[71], a collaborative international study sampled from cooperating gender identity clinics in Belgium, the Netherlands, Italy and Norway. Additionally, three studies (16%) used convenience sampling in the United States, Germany/Switzerland and Australia, including online sampling. Four studies (21%) included cisgender control groups, two studies (11%) included transgender people not undergoing gender-affirming hormone therapy as controls, and one study (5%) included both types of control groups. While 18 (95%) of these studies included indicators of well-being (the majority of studies focused on depression, anxiety and psychological distress), only six (32%) included measures relevant to self-mastery (typically measures of anger), and only three (16%) included measures relevant to interpersonal functioning.

**Feminizing hormone therapy.** *Well-being.* The most consistent evidence for changes in psychosocial functioning after feminizing hormone therapy included reductions in psychological distress in four studies[72–75] and reductions in depressive symptoms in three studies[72,73,76]. One study[77], with low risk of bias, a sample of 17 youth (ages 9–25 years old) on feminizing hormone therapy in the United States (with no power analysis), and follow-up at 6 and 12 months, found no statistically significant effect on change in depressive symptoms from 0 to 12 months on the Center for Epidemiologic Studies Depression Scale (CESD-R) or the PHQ-9 Modified for Teens, although the authors noted that both effect sizes were 'notably large' in the direction of reduced depressive symptoms. Another study[78], with high risk of bias, with 14 people on feminizing hormone therapy in the Netherlands, and no power analysis, also found no statistically significant differences after 8 weeks on the Self-Rating Depression Scale (SDS).

For anxiety, two studies[76,79] conducted in the United Kingdom, with moderate risk of bias, both including power analyses, and samples of 59 and 95 people, respectively, on feminizing hormone therapy, showed no significant evidence of differences after 12 (ref. 79) and 18 (ref. 76) months on the Hospital Anxiety and Depression Scale (HADS-A). The study[78] in the Netherlands, with high risk of bias, also found no statistically significant difference after 8 weeks on the Spielberger Trait Anxiety Inventory (STAI). However, another study[72] in Italy, with moderate risk of bias, found significant reductions in anxiety after 12 months on the Self-Rating Anxiety Scale (SAS) in a sample of 78 people on feminizing hormone therapy.

Quality of life was found to be greater after feminizing hormone therapy in two studies[80,81]. However, the study[77] with low risk of bias, but a sample of only 17 youth on feminizing hormone therapy in the United States, and follow-up at 6 and 12 months, showed no statistically

significant difference on the Pediatric Quality of Life and Enjoyment Scale (PQLES-SF); another study[82], with moderate risk of bias and follow-up at 3 and 6 months, also found no significant difference on the Short Form Health Survey (SF-36) in a sample of 35 people on feminizing hormone therapy in Australia. Neither of these studies included a power analysis.

For affect, one study[83], with low risk of bias, from the ENIGI cohort, found no change in negative affect but a decrease in positive affect after feminizing hormone therapy in a 3-year follow-up. The decrease in positive affect reported in this study emerged in the first 3 months and then stayed stable over repeated follow-ups over 3 years, only returning to levels statistically non-significantly different from baseline at the final time point. One study[35], with moderate risk of bias, conducted in the Netherlands with a sample of 47 people on feminizing hormone therapy and follow-up after about 3 months, found evidence of increased affect intensity and increased emotional expressiveness, while another more recent study[75], with low risk of bias, conducted in Italy, found reductions in alexithymia after 12 months for 24 people on feminizing hormone therapy.

One study[84], with low risk of bias, conducted in the Netherlands with a sample of 21 youth (ages 11–27 years old) who had been on feminizing hormone therapy for at least 6 months, showed greater self-esteem.

*Self-mastery.* For self-mastery, three studies[35,78,85] found no significant differences in anger intensity—with the more recent study, utilizing the ENIGI cohort and a 3-year follow-up, having lower risk of bias, but none including a power analysis. However, one study[35], with moderate risk of bias, a sample of 47 people on feminizing hormone therapy and follow-up after about 3 months showed increased anger readiness. Similarly, another study[36], also conducted in the Netherlands in the early 1990s, with low risk of bias, but a sample of only 15 transgender women and no power analysis, with follow-up after about 3 months, showed increased anger proneness on the Anger Expression Scale (AX) and Anger Situation Questionnaire (ASQ) after feminizing hormone therapy.

One more recent study from the Netherlands[84], with low risk of bias but no power analysis, found no significant difference in behavioural conduct problems after being on feminizing hormone therapy for at least 6 months, in a sample of 21 youth.

*Interpersonal functioning.* One study conducted in Italy[75], with low risk of bias and 24 people on feminizing hormone therapy, found reductions in social anxiety after 12 months, while another study[84], with low risk of bias and no power analysis, conducted in the Netherlands with a sample of 21 youth, found no significant differences in close friendship or social acceptance after at least 6 months on feminizing hormone therapy.

*Longitudinal summary.* Taken together, these prospective cohort studies suggest that feminizing hormone therapy reduces psychological distress and depressive symptoms as well as potentially improves quality of life, all indicators of improvements in well-being. For affect, one high-quality study showed potential reductions in positive affect over the course of a 3-year follow-up, but no differences in negative affect, while other studies suggested increased emotional expressiveness and affect intensity as well as reduced alexithymia. In terms of self-mastery, effects on anger were mixed, with the highest quality study finding no differences but smaller and less recent studies finding increased anger readiness and proneness after 3 months, suggesting that perhaps time course does matter to an extent, with changes evident earlier (that is, in the first 3 months of hormone therapy). Evidence for changes in interpersonal functioning came from only two studies and was inconclusive.

**Masculinizing hormone therapy.** *Well-being.* As with feminizing hormone therapy, the most consistent evidence for changes

in psychosocial functioning after masculinizing hormone therapy involved reductions in psychological distress in five studies[72–75,86] and depressive symptoms in three (refs. 72,73,76). Similar to feminizing hormone therapy, the same study[77], from the United States, with low risk of bias, found no statistically significant effect of masculinizing hormone therapy on depressive symptoms in a sample of 33 youth from 0 to 12 months' follow-up on the CESD-R or PHQ-9 Modified for Teens, with the same caveat that both effect sizes were 'notably large' in the direction of reduced depressive symptoms.

For anxiety, two studies[76,79] in the United Kingdom, with moderate risk of bias, showed no significant evidence of differences after 12 (ref. 79) and 18 (ref. 76) months on the Hospital Anxiety and Depression Scale (HADS-A), but both included power analyses reporting adequate statistical power, with samples of 59 and 83 people on masculinizing hormone therapy, respectively. However, one study[72], with moderate risk of bias, found significant reductions in anxiety after 12 months on the SAS in a sample of 29 people on masculinizing hormone therapy in Italy.

Two studies[80,82] reported greater quality of life for those on masculinizing hormone therapy after 3 (refs. 80,82) and 6 (ref. 82) months, but two other studies[77,81] found no significant differences: one study[77], with low risk of bias, a sample of 33 youth on masculinizing hormone therapy, in the United States and follow-up at 6 and 12 months, showed no statistically significant difference on the Pediatric Quality of Life and Enjoyment Scale (PQLES-SF); and another study[81], with moderate risk of bias, a sample of 27 people on masculinizing hormone therapy, in Italy and follow-up at 12 months, showed no statistically significant difference on the World Health Organization Quality of Life Questionnaire (WHOQOL-100); neither study included a power analysis.

Interestingly, measures of affect showed evidence of affective dampening after masculinizing hormone therapy, including less positive and negative affect in the ENIGI cohort[83], as well as less affect intensity, but not a significant difference in emotional expressiveness, in one study[35], with moderate risk of bias, conducted in the Netherlands with 54 people on masculinizing hormone therapy and follow-up after 14 weeks. One study[87], with moderate risk of bias, conducted in Germany and Switzerland, with 23 people on masculinizing hormone therapy and follow-up at 3 and 6 months, found reductions in neuroticism, while another study[75], with low risk of bias conducted in Italy, found reductions in alexithymia after 12 months for 38 people on masculinizing hormone therapy.

One study[84], with low risk of bias, conducted in the Netherlands with a sample of 49 youths, showed greater self-esteem after at least 6 months on masculinizing hormone therapy.

*Self-mastery.* Three studies[35,36,88] showed increases in anger expression[88], anger readiness[35] and anger proneness[36], all within 3–7 months of commencing masculinizing hormone therapy. However, two studies[35,85] found no significant differences in anger intensity after masculinizing hormone therapy, including in the ENIGI cohort[85] (although this study did report a trend toward increased anger intensity only after 3 months, but not at any other follow-up out to 36 months, in those on masculinizing hormone therapy).

Another study[84], from the Netherlands, with low risk of bias, found fewer behavioural conduct problems after at least 6 months on masculinizing hormone therapy in a sample of 49 youths.

*Interpersonal functioning.* For interpersonal functioning, one study[75], with low risk of bias, with 38 people on masculinizing hormone therapy, conducted in Italy, found reductions in social anxiety after 12 months, while another study[87], with moderate risk of bias, conducted in Germany and Switzerland, with 23 people on masculinizing hormone therapy, found increases in extraversion and agreeableness after 3 and 6 months of follow-up.

One study[84], with low risk of bias, conducted in the Netherlands, found no significant differences in close friendship or social acceptance

after at least 6 months in a sample of 49 youth on masculinizing hormone therapy, again with no power analysis.

*Longitudinal summary.* Taken together, these prospective cohort studies tended to show that masculinizing hormone therapy reduced psychological distress and depressive symptoms. Effects on anxiety and quality of life were mixed. Notably, some studies suggested a dampening in affective experiences after masculinizing hormone therapy, although it is unclear whether this might indicate improvements or decrements in well-being given that this encompassed dampening of indicators of both positive and negative affect. For self-mastery, some studies indicated increased anger readiness and expression as well as proneness to anger, potential indicators of decrements in self-mastery, although this was not confirmed by increased anger intensity, suggesting that the emotion itself might not be affected, but rather the willingness to show/express it. Limited evidence on interpersonal functioning suggested improvements here too.

### Adjustment for potential confounders

Cross-sectional studies provide limited evidence for changes in psychosocial functioning after gender-affirming hormone therapy due to a lack of comparison of outcomes across time, which is required to reflect change. Prospective cohort studies are the gold-standard in terms of evidence for change; however, these studies are also vulnerable to risk of bias due to confounding factors. This is particularly important in this case because as people undergo gender transition, many physical and social changes occur that can, by themselves, explain the results discussed in this review. Despite pre-/post-hormone therapy designs, prospective cohort studies showed substantial risk of bias related to confounding. For example, gender transition can markedly improve body image, which by itself can improve psychosocial functioning[89]. At the same time, for some people transitioning is accompanied by a substantial amount of exposure to stigma, and even aggression, thereby negatively affecting at least some indicators of well-being (for example, anxiety) or interpersonal functioning (for example, social phobia).

Table 1 lists the various potential confounders adjusted for across the included studies. Notably, measures of body image and gender-affirming surgeries were most commonly included, but these critical confounders were still only adjusted for in about one-third of the quantitative studies. Other potentially critical confounders, such as gender affirmation by others and social stigma[90], were only adjusted for in two studies each. Given these limited attempts to adjust quantitative estimates for plausible, or even known, confounders, the extant quantitative literature cannot be conclusive in terms of biological versus psychological or sociocultural pathways by which gender-affirming hormone therapy might influence psychosocial outcomes. This remains a substantial limitation of this body of work.

## Discussion

The current systematic review highlights the state of the science concerning the potential effects of gender-affirming hormone therapy on psychosocial functioning for transgender people (see Fig. 2 for a summary of the main findings of the review). The most consistent evidence across qualitative and quantitative studies, both cross-sectional and prospective cohorts, is that gender-affirming hormone therapy reduces depressive symptoms and psychological distress, consistent with results of previous systematic reviews[37–40]. There was also some evidence of potential reduction in general anxiety among those on masculinizing hormone therapy; however, this was primarily demonstrated in cross-sectional studies and not yet substantiated in prospective cohort designs. Notably, these changes all reflect reductions in distress rather than direct increases in positive states, suggesting that gender-affirming hormones may improve well-being primarily by helping to eliminate gender dysphoria, whether through improved body image or other relevant pathways, or even simply increasing an

Feminizing hormone therapy

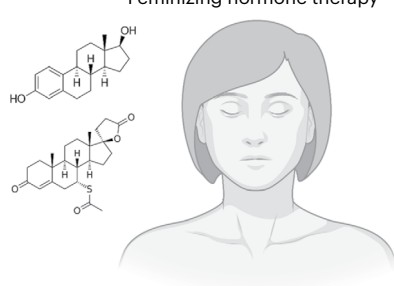

**Stronger evidence**
↓ Psychological distress
↓ Depressive symptoms

**Weaker evidence**
↑ Quality of life
↑ Emotional imbalances
↑ Emotional expressiveness

Masculinizing hormone therapy

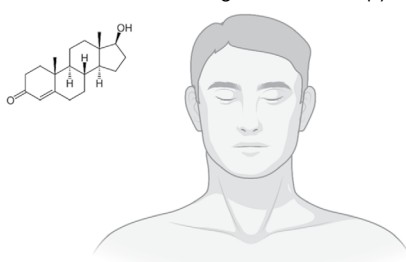

**Stronger evidence**
↓ Psychological distress
↓ Depressive symptoms

**Weaker evidence**
↓ General anxiety symptoms
↓ Social anxiety symptoms
↑ Quality of life
↑ Affective dampening
↑ Anger expression

**Fig. 2 | Summary of evidence for psychosocial effects of gender-affirming hormone therapy.** Stronger and weaker evidence for psychosocial effects of gender-affirming hormone therapy is summarized for those on feminizing and masculinizing hormone therapy separately. Chemical symbols represent oestradiol (top) and spironolactone (bottom) for a common approach to feminizing hormone therapy, and testosterone for a common approach to masculinizing hormone therapy. Figure created with BioRender.com.

individual's sense of control and autonomy over one's body and gender expression.

Evidence for effects of hormone therapy on quality of life and affect in the current review is inconsistent, but tends to point in the direction of improvements. However, there was some quantitative evidence of affective dampening among those on masculinizing hormone therapy, which may be related to the restricted range of emotions described by participants on masculinizing hormone therapy in qualitative work. Relatedly, in the current review we found that participants on feminizing hormone therapy sometimes described mood swings and emotional imbalances, but also gave greater insight into their emotions and increased emotional expressiveness. Evidence from an ad hoc questionnaire of side-effects included in the ENIGI cohort[91,92] further substantiates these differences in emotionality between those on masculinizing and feminizing hormone therapy. Given these suggestive results, future research should continue to probe the duration of any potential changes in emotionality and whether they are experienced as distressing, or even pleasing, by transgender people at any point in time.

Self-mastery was the aspect of psychosocial functioning with the most inconclusive results in the current review. We found that earlier studies, conducted in the 1990s and early 2000s, hinted at decreased self-mastery in the form of greater readiness to act on anger, both for those on masculinizing and feminizing hormone therapy. This was also confirmed in a more recent study with people on masculinizing hormone therapy showing greater anger expression[88], but not examined recently in those on feminizing hormone therapy. However, anger intensity does not seem to increase for those on either masculinizing or feminizing hormone therapy. These results are consistent with a recent systematic review of effects of testosterone therapy on aggression in transgender men[93]. Importantly, while self-mastery goes beyond anger expression, no studies have examined other elements of self-mastery,

such as self-control or impulsivity, with only one study[84] investigating behavioural conduct in transgender youth. This remains an important avenue for future investigation.

Evidence from the current review for improvements in the domain of interpersonal functioning is limited and inconclusive, but hints toward positive change, particularly for those on masculinizing hormone therapy, most commonly in the form of reduced social anxiety. Given the dearth of research on interpersonal functioning in prospective cohort studies, future longitudinal research on gender-affirming hormone therapy should aim to include relevant measures, such as measures of loneliness (for example, UCLA Loneliness Scale), trust (for example, Rempel and Holmes Trust Scale), attachment styles (for example, Experiences in Close Relationships Scale), relationship satisfaction (for example, Couples Satisfaction Index) and other forms of social functioning.

Notably, the juxtaposition of increased anger readiness and decreased social anxiety specifically for those on masculinizing hormone therapy is interesting in that it may point toward a causal role for testosterone in these changes. On average, cisgender women show lower rates of aggression[94] and higher rates of social anxiety[95] relative to cisgender men. This inversion of traits could therefore be driven in part by the effects of testosterone, pushing those on masculinizing hormone therapy toward more assertive and self-confident behaviour in social interactions[96,97]. Future work is necessary to confirm this possibility and attempt to parse biological and sociocultural pathways.

It is interesting to note that these changes, along with others, might also reflect endorsement of gendered stereotypes pertaining to psychosocial functioning. For example, the affective dampening reported in some studies among participants on masculinizing hormone therapy is in line with gender role expectations that men should not experience or express 'too much' emotion, while greater anger expression could be related to the fact that this specific emotion is an exception to, and reversal of, this norm[98]. Among participants on feminizing hormone therapy, reports of mood swings and emotional imbalances may be in line with core stereotypes of women[99]. If these changes, identified on self-report measures, do actually reflect gender-stereotyped expectations on the part of participants rather than objective changes in psychosocial functioning, other types of tasks that do not rely on self-report exclusively (for example, behavioural tasks) may prove useful in unpacking these preliminary findings.

Across all outcomes, consideration of study quality, measures, sample size and other factors is necessary. Generally, studies were at risk of bias due to confounding and many included small samples, with consequent low statistical power, to detect changes in psychosocial functioning, along with frequent lack of correction for multiple tests to reduce family-wise error rates. Furthermore, while the variety of measures used in the literature is in some ways a strength (that is, in terms of generalizability of effects), it is difficult to compare inconsistent results across studies when they may be a product of the specific measures, or operationalizations, used in each study.

In quantitative studies, including cross-sectional and prospective cohort studies, there is a need for research with adequate control groups, potentially composed of matched cisgender people as well as transgender people not wishing to undergo gender-affirming hormone therapy, as well as those waitlisted for treatment[100]. Identifying appropriate control groups is no easy task, given that these might differ depending on the outcome examined. For example, transgender people waiting for hormone therapy might be particularly distressed because they feel that what they need is still far away. So, with regard to well-being, perhaps people who do not want hormone therapy are a better comparison. However, it is of course possible that there are already differences in psychosocial functioning between people who do and do not seek hormone therapy to affirm their gender, meaning that studies utilizing both types of control groups and following changes in each over time may be most appropriate.

One important issue that has not been systematically considered in the literature on gender-affirming hormone therapy is the importance of timing. Findings from the current review point toward a potentially critical role of time course in shaping effects on various facets of psychosocial functioning (for example, there is some evidence from the current review, as well as another recent review[93], that increases in anger expression among those on masculinizing hormone therapy may be short-lived, appearing only in the first 3–6 months after beginning hormone therapy and then reducing to baseline levels over time). Relatedly, physical changes resulting from gender-affirming hormone therapy can vary in onset and duration; for example, for those on feminizing hormone therapy, breast development may begin at about 3 months but only reach desired levels after about 3 years[101]. It is also unclear whether specific psychosocial changes may reverse if gender-affirming hormone therapy is ceased[102]. Developmental timing is also a critical consideration, particularly for transgender children and adolescents, for whom puberty may induce changes that are more difficult to modify later in life or are even irreversible[103]. Future research examining the effects of hormone therapy on psychosocial outcomes needs to explicitly take time into account in all of these different ways.

Furthermore, in all of these studies, participants were transgender people who volunteered to participate in a study, so results might depend on how the study's aims were introduced to them in interaction with the agenda they might have wanted to push forward. For example, studies stating their goals were to examine effects of hormone therapy might have attracted participants who wanted to demonstrate its benefits (which, in addition, may have been understood to be better demonstrated by expressing gender stereotypical affective responses, for example), whereas studies claiming their aim was on understanding well-being among transgender people might have attracted more participants who wanted to voice negative experiences. These details might in fact explain some of the contradictions found and point to the need to consider these more psychosocial aspects of transition with greater care when conducting this kind of research.

In addition, without a better understanding of what transgender people wish to achieve with their transition, it is hard to know exactly whether a particular change is positive or negative. For example, increases in anger in those on masculinizing hormone therapy might at first seem negative, but they might be experienced positively if the person in question sees this as confirming their masculinity. At the same time, outcomes need to be understood by reference to what transgender people think medical providers wish to achieve, or even what medical providers tell them about desirable or expected effects. Given the authority of medical providers to determine treatment courses, some transgender people might wish to report effects that are in line with what is expected by providers (again, potentially gender stereotypical responses) so as to ensure the continuation of treatment. Alternatively, people might have been given expectations that are thwarted and low well-being might refer more to disappointment than to biological effects of hormones.

An important starting point in understanding potential psychosocial implications of gender-affirming hormone therapy is indeed to listen to the voices and experiences of transgender people. There have been calls among scholars to prioritize empowerment of transgender people and communities in relevant research, especially by including their voices throughout the research process[104]. Despite the importance of this area of research and the logic of querying transgender people's lived experiences as they relate to gender-affirming hormone usage, very few studies have been conducted on this topic. Furthermore, only two of the six studies identified here included people on masculinizing hormone therapy in the sample. Therefore, the voices of transmasculine people, as well as experiences on masculinizing hormones, are almost entirely absent from the literature.

While the aim of the current review was to assess the evidence for changes in psychosocial functioning following gender-affirming

hormone therapy, this topic may be inherently linked to potential changes in cognitive functioning as well. Other recent systematic reviews[40,105] have hinted at a link between gender-affirming hormone therapy and cognitive functioning changes, including improved visuospatial ability among those on masculinizing hormone therapy. Such cognitive changes, potentially driven by changes in brain structure or function[29,106], may overlap with how the brain processes social and emotional stimuli, influencing psychosocial functioning for transgender people after gender-affirming hormone therapy. Furthermore, where cognitive changes align with gender stereotypes, they may be felt to confirm successful gender transition and, thereby, enhance well-being.

There is evidence that gender-affirming hormone therapy results in improved psychosocial functioning for transgender people, primarily improved well-being. Changes in self-mastery and interpersonal functioning are more ambiguous—with patterns across these dimensions that may diverge for those on masculinizing versus feminizing hormone therapies. Given the paramount importance of social relationships to health[20,21], further high-quality evidence for psychosocial effects of gender-affirming hormone therapy (for example, on self-mastery and interpersonal functioning) is vital to ensuring health equity for transgender people. Attempts to limit access to gender-affirming care, including hormone therapy, have sometimes relied on a lack of scientific evidence for various outcomes[41,42], but the current review points toward improvements in overall well-being, particularly in the form of reduced distress, implying that any potential risks to other facets of psychosocial functioning, which are likely limited, are outweighed by the benefits of these vital treatments for transgender health. Continuing carefully conducted and executed research on this topic will be essential to mitigating any potential risks and promoting transgender health across countries in future.

## Methods

### Eligibility criteria

All empirical research published or in press by May 2022 was considered for inclusion in the current systematic review. To be deemed eligible, research needed to (1) include transgender participants who had previously used or were currently using gender-affirming hormone therapy (not including studies focused exclusively on puberty suppression without further gender-affirming hormone therapy); (2) evaluate at least one psychosocial outcome as broadly defined in the introduction—when multiple relevant outcomes were presented, all were included, but we chose to include total scale scores over subscales when both were presented separately in one paper; (3) encompass normal levels of functioning—that is, we chose not to include studies that examined changes in clinical psychiatric diagnoses; (4) for quantitative research, provide a relevant comparison either within-person before and after gender-affirming hormone therapy, or between-person comparing transgender people who had previously or were currently using gender-affirming hormone therapy to transgender people who had not yet used gender-affirming hormone therapy, or who were on a waitlist to receive gender-affirming hormone therapy; (5) for quantitative research, include psychometrically validated quantitative measures rather than ad hoc self-reports of symptoms or review of medical chart notes—although we chose to exclude studies utilizing the Minnesota Multiphasic Personality Inventory or similar measures involving sex norms, as interpretation of these sex-standardized scales is problematic in transgender samples given that the sex norms have been validated with exclusively cisgender samples;[107] and (6) for qualitative research, include a formal and systematic thematic analysis rather than a cursory description of observations from text.

### Search strategy and study selection

We followed PRISMA guidelines when carrying out the search strategy for this review[108]. Key terms (hormones, hormone replacement therapy, testosterone replacement therapy, oestrogen replacement

therapy, gender affirmation, gender-affirming, health, well-being, psychosocial functioning, transgender, non-binary, gender diverse, gender fluid, transmasculine, transfeminine, transsexual) were identified from a number of articles[37,39,40,109,110] on the topic along with MeSH terms. PubMed, PsycNet and Web of Science databases were searched between the years 1980 and 2022. This time period was selected due to the relative recency of papers that explore this topic as highlighted in similar prior reviews, where the dates of individual papers did not stretch past 40 years before the search despite the wider date range employed[50,110]. We chose to restrict our search to the published academic literature as (1) critics of gender-affirming hormone therapy have relied on arguments for lack of consensus in the published academic literature to attempt to restrict healthcare access for transgender people, therefore a comprehensive and systematic review of this work is necessary to adjudicate such claims and (2) differences in the types of grey literature available (that is, more cross-sectional studies relative to prospective cohorts) could lead to biases in the conclusions drawn between these different sections of the review. Terms for transgender people (for example, transgender, non-binary, gender diverse, gender fluid, transmasculine, transfeminine, transsexual and gender dysphoria) were searched using the OR function and were combined with terms related to psychosocial outcomes (for example, mood, anxiety, depression, self-esteem), as well as hormone usage (for example, hormones, cross sex hormone therapy, testosterone replacement therapy, feminizing, masculinizing)—(see Supplementary Tables 3 and 4 for tables containing search history and final search terms). Additionally, the reference lists of selected articles as well as the 'similar readings' function in PubMed were searched to identify any further possible relevant papers (see Fig. 1 for the study inclusion flowchart and Supplementary Table 5 for a description of all studies included in the review).

## Data availability

No specific datasets were generated for the current systematic review. A table containing coding of risk-of-bias for all studies in the current review can be found in the Supplementary Information.

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

## Acknowledgements

This study was funded by the European Union (ERC-StG 101042028 to D.M.D.). Views and opinions expressed are, however, those of the authors only and do not necessarily reflect those of the European Union or the European Research Council. Neither the European Union nor the granting authority can be held responsible for them. The funders had no role in the research design, decision to publish or preparation of the manuscript.

## Author contributions

D.M.D. conceptualized the research. D.M.D. and T.O.G.L. performed the systematic review. T.O.G.L. coded studies with supervision from D.M.D. and M.B. All authors drafted and reviewed the final manuscript.

## Competing interests

The authors declare no competing interests.

## Additional information

**Correspondence and requests for materials** should be addressed to David Matthew Doyle.

