## [Peer Review File · Nature Human Behaviour]

Peer Review Information

Journal: Nature Human Behaviour

Manuscript Title: A Systematic Review of Psychosocial Functioning Changes after Gender-Affirming Hormone Therapy among Transgender People

Corresponding author name(s): David Matthew Doyle

Reviewer Comments & Decisions:

Decision Letter, initial version:

16th May 2022

Dear Dr Doyle,

Thank you once again for your manuscript, entitled "A Systematic Review of Psychosocial Functioning Changes after Gender-Affirming Hormone Therapy among Transgender People", and for your patience during the peer review process.

Your Article has now been evaluated by 3 referees. You will see from their comments copied below that, although they find your work of potential interest, they have raised quite substantial concerns. In light of these comments, we cannot accept the manuscript for publication, but would be interested in considering a revised version if you are willing and able to fully address reviewer and editorial concerns.

We hope you will find the referees' comments useful as you decide how to proceed. If you wish to submit a substantially revised manuscript, please bear in mind that we will be reluctant to approach the referees again in the absence of major revisions. We are committed to providing a fair and constructive peer-review process. Do not hesitate to contact us if there are specific requests from the reviewers that you believe are technically impossible or unlikely to yield a meaningful outcome.

To guide the scope of the revisions, the editors discuss the referee reports in detail within the team, including with the chief editor, with a view to (1) identifying key priorities that should be addressed in revision and (2) overruling referee requests that are deemed beyond the scope of the current study. We hope that you will find the prioritised set of referee points to be useful when revising your study. Please do not hesitate to get in touch if you would like to discuss these issues further.

1. It is very important to ensure that your review is up to date and includes all of the recently published literature, as Reviewer 1 points out. We ask that you update your search to include papers up to the time of receipt of this decision letter. When updating your search, please do take into

consideration and justify or otherwise amend your choice to apparently limit your search to the published literature only.

2. Reviewers 2 and 3 raise some important methodological concerns, as well as concerns related to the quality of the included studies. We ask that you address these very carefully and formally evaluate heterogeneity and risk of bias of the studies included in your meta-analysis. Please also submit an up-to-date version of your PRISMA checklist with your revised manuscript.

3. We urge you to follow Reviewer 3's recommendation to engage stakeholders from the transgender community to ensure that your manuscript presents and discusses the context and synthesized literature appropriately and sensitively.

4. Please make sure that all claims are appropriately supported either by relevant citations or meta-analytic evidence, as all three reviewers request.

If you wish to submit a suitably revised manuscript we would hope to receive it within 4 months. I would be grateful if you could contact us as soon as possible if you foresee difficulties with meeting this target resubmission date.

- Include a "Response to the editors and reviewers" document detailing, point-by-point, how you addressed each editor and referee comment. If no action was taken to address a point, you must provide a compelling argument. When formatting this document, please respond to each reviewer comment individually, including the full text of the reviewer comment verbatim followed by your response to the individual point. This response will be used by the editors to evaluate your revision and sent back to the reviewers along with the revised manuscript.
- Highlight all changes made to your manuscript or provide us with a version that tracks changes.

[REDACTED]

Thank you for the opportunity to review your work. Please do not hesitate to contact me if you have any questions or would like to discuss the required revisions further.

Sincerely,

Samantha Antusch

Samantha Antusch, PhD
Editor
Nature Human Behaviour

Reviewer expertise:

Reviewer #1: mental health and psychosocial outcomes in transgender individuals

Reviewer #2: gender-affirming therapy for transgender individuals ; mental health and psychosocial outcomes in transgender individuals ; systematic review

Reviewer #3: gender-affirming therapy for transgender individuals ; mental health and psychosocial outcomes in transgender individuals

REVIEWER COMMENTS:

Reviewer #1:

Remarks to the Author:

Thank you for the opportunity to review "A Systematic Review of Psychosocial Functioning Changes after Gender-Affirming Hormone Therapy among Transgender People." I appreciate the importance of summarizing the state of the literature regarding gender affirming hormones and specifically related to psychosocial functioning. The authors do a nice job of summarizing the importance of a systematic review such as this one. I agree that this is an important topic area. There are some major areas of revision that would be required, in my opinion, to be considered for publication. First, there have been additional studies that have been published on this topic in the last year and the authors should update accordingly. Also, there were articles that were missed and the authors should do additional searches to ensure they have covered more of the literature. Finally, the authors need to be mindful of making conclusions or sweeping comments about findings without having meta-analytic data to back up the assertions.

In general, instead of using binary language of transmasculine and transfeminine, I recommend focusing on the naming the type of hormones that people would receive, for example: masculinizing hormones or feminizing hormones, given the number of nonbinary, agender, genderfluid, etc. people who will likely be included in your review.

Abstract:

I recommend including nonbinary populations in the abstract and in the review, given the number of nonbinary people who access gender affirming hormones. See line 16 where only transgender people are mentioned.

The authors mention 32 papers—I'm curious as a reader at this point if this means journal articles, book chapters, and nonpublished papers (dissertations) or if the authors mean journal articles only. See line 16.

On line 20, what does "more mixed" mean—more than what?

Introduction:

The commentary on why studying psychosocial functioning as an outcome of hormone therapy is compelling and demonstrates that importance of this study. As a minor note, the word "but" on line 44 almost downplays your argument—I would just simply state the additional outcome of psychosocial states and social interactions.

Are the studies starting on line 60 only focused on cisgender populations?

I am always a bit concerned in literature reviews when disparities are discussed without the theoretical explanation for why they exist in the first place (e.g., minority stress causing significant levels of social anxiety). I recommend at least contextualizing the disparities that are discussed starting on line 62.

I'm curious about how relevant the Bos et al. 2010 study is, given that cis women were only given sublingual testosterone on two separate days and then given facial trustworthiness evaluations—I would recommend that citations that follow people who take hormones for a period of time be included as evidence in this study, given the arguments you make throughout your manuscript.

I'm wondering where the citation is for this assertion: "Currently, transgender people are routinely prescribed such biological interventions without being made sufficiently aware of potential psychosocial implications (and available counselling options to support positive adjustment), potentially risking social disruption and, therefore, poorer health and well-being across the life course." See line 98.

It would be helpful to specify the aims further—you note on line 114: "That is, we aimed to understand not only what effects hormone therapy might have on psychosocial functioning, but also whether these could be unambiguously and directly related to the hormonal changes, or whether existing evidence does not separate this from the various other changes that are associated with gender identity transition." It might be useful to mention how you are able to determine the second part of this aim or specify further.

Method:

The data for this study are a year old—most journals will not publish meta analysis or systematic reviews unless the data are analyzed within 3 months of submissions. I recommend this gold standard for this particular review as well.

What does this mean on line 126: "encompass normal levels of functioning/symptomatology and not focus exclusively on clinical diagnoses/thresholds"

Were only journal articles used in this review? And if so, why? And what does that mean for the file

drawer effect?

It seems like being more specific in the qualitative section would be beneficial to this review. Again, I suggest having two sections—one regarding feminizing hormones and one regarding masculinizing hormones. Given the second aim of the study, I'm wondering if the authors looked into determining how findings could delve into the complexity of outcomes related to hormone therapy. Also, I wonder if emailing the authors for the Riggs study to determine if there were quotes related to individuals taking masculinizing hormones are available for the review?

In reading the cross-sectional section, I'm wondering if there was a cap at the timeframe for when studies were published? It seems that the process of administering hormone therapy has changed over the last several decades—it might be worthwhile to note or comment on some of the older studies vs the new ones—the Blanchard 1983 study stands out compared to the other relatively newer ones.

But also, as I read through the review, I noticed some studies were missed, for example this one and subsequent ones out of this dataset:

Keo-Meier, C. L., Herman, L. I., Reisner, S. L., Pardo, S. T., Sharp, C., & Babcock, J. C. (2015). Testosterone treatment and MMPI-2 improvement in transgender men: A prospective controlled study. *Journal of consulting and clinical psychology*, 83(1), 143.

Discussion:

This seems like a leap for a systematic review, see line 417: "Interestingly, this happens irrespective of whether the treatment is feminizing or masculinizing, which might be taken to suggest that these well-being improvements are not necessarily directly due to hormonal changes, but to the positive effects of transition on body image (or other confounders), which in turn improves well-being (or even simply a reflection of the positive sense that having control and autonomy over one's body and gender expression brings)." Unless the authors are willing to analyze the data in aggregate, it seems impossible to make this assertion from systematic review. There are confounding variables for every medical (and psychological) study and this assertion could be made for pretty much anything—without actually having data, it seems impossible to know what the direct effects of treatments might be.

I would suggest that the authors do a deeper dive into the section starting on page 422. It might be useful to pose additional questions about blunted affect as a result of testosterone—does this cause distress to the people experiencing it? Similar with mood swings and emotional imbalances—it seems like posing research questions around these would be useful—are these clinical levels emotional changes or are they changes in people actually experiencing their emotions more deeply for the first time or in a way they haven't in a while?

On line 445, it would be helpful to mention specific measures that researchers could use.

Reviewer #2:
Remarks to the Author:

I appreciate the opportunity to review this systematic review on the important topic of psychosocial changes associated with gender affirming hormone therapy. Overall, the manuscript addresses a topic that has received insufficient attention and effectively summarizes what is known and where more data are needed. Findings will be informative for people who provide care for transgender clients as well as for researchers. That said, there are several areas that would benefit from strengthening. Below I outline my recommendations by section of the manuscript:

ABSTRACT

The phrase, "...with hints toward positive change.." is colloquial. Scientific language such as "trend toward positive change" is more appropriate.

INTRODUCTION

The first paragraph is missing several citations and some of the citations provided are quite dated. For example, one reference referred to data from 2010-2014. Citations are needed to support the following statements: (1) "most transgender people continue to use some dosage of gender-affirming hormones throughout their lives" (2) " hormones also affect psychological states and social interactions" (3)"large and growing prevalence of gender-affirming hormone therapy across countries"

The statement at the end of the first paragraph that gaps in knowledge limit fully informed consent is inaccurate. Informed consent processes require sharing with the client what is known and unknown about a procedure or treatment. It does not require that everything be known in order to be fully informed consent.

The Testa 2017 citation at the end of the second paragraph does not support the claim that suicide risk in trans people is linked to disruption in their social lives. The cited manuscript links minority stressors such as rejection, non-affirmation, victimization, and discrimination, to suicide risk. If this is what is meant by "disruption in social lives" please be specific about that in the manuscript.

The third paragraph provides definitions of terms that have already been used throughout the first 2 paragraph. I recommend moving it up to be the first paragraph.

The 5th paragraph makes a strong claim that essentially insults all prescribers of gender affirming hormones: "Currently, transgender people are routinely prescribed such biological interventions without being made sufficiently aware of potential psychosocial implications (and available counselling options to support positive adjustment), potentially risking social disruption and, therefore, poorer health and well-being across the life course." Please provide a citation supporting this claim that prescribers do not make transgender people aware of known psychosocial implications. The fact that this is not described in current guidelines does not mean that it does not routinely happen.

In the final sentence of the Introduction, I recommend removing "identity" from the sentence. Trans people on hormone therapy undergo gender transition (of their bodies) but not a transition of their identity.

METHODS

Section 2.2: Who is "they" in the first sentence?

Please provide an exact list of the terms used for the search, including MESH terms, rather than supplying only examples. A table of search terms may be best for this.

RESULTS

Instead of general statements such as "large" and "small" sample sizes for the quantitative studies, please provide the exact sample sizes of the studies in the text when discussing their findings. Please note if any of the studies provided a power analysis, particularly those with null findings that may have been related to inadequate power to detect a significant difference.

In section 3.2.1.1.1, the penultimate sentence states that comparisons between studies were "unclear" due to use of different measures. However, it seems more appropriate to state that comparisons across studies with different measures is not valid.

I recommend replacing potentially offensive term, "black market" with a term like "informal sector" or "from unlicensed providers."

DISCUSSION

The first sentence refers to "state of the art"; however "state of the science" is more appropriate in this context.

The first sentence of the 3rd paragraph refers to "ambiguous results"; however, "inconsistent" or "inconclusive" would be a more appropriate term.

Citations are missing for multiple statements of fact in the third paragraph. The second sentence refers to multiple earlier studies but does not cite any studies at all. The next sentence refers to confirmation in a more recent study but does not cite any study. It's not clear if this sentence: "However, anger intensity does not seem to increase for those on either masculinizing or feminizing hormone therapy." refers to findings from this review or to earlier studies; if it refers to earlier studies, a citation is needed. In the following paragraph, no citation is provided for this statement: "Evidence for improvements in this domain is limited and inconclusive, but hints toward positive change, once again for those on both feminizing and masculinizing hormone therapy (most commonly in the form of reduced social anxiety)."

In the 5th paragraph, is the author indicating that self-report bias is based on internalized stereotypes? If so, please state this clearly. If not, it's unclear how self-report bias is related to the

discussion of stereotypes earlier in the paragraph.

In the 7th paragraph, please indicate the source for this quote, "are where they want to be." If this is not a quote, then I recommend replacing this colloquialism with the statement that is in parentheses after the quotation marks.

The authors appropriately note that waiting for hormone therapy may cause distress that could serve as a confounder when comparing those waiting with those currently on gender affirming hormone therapy. They fail to note that there may ALSO be significant psychosocial differences (confounders) between trans people who are on hormones and those who do not want hormones, making that comparison group also potentially problematic.

In the 8th paragraph, I recommend replacing "in their full glory" with more professional language.

In the 10th paragraph, it's not clear what is meant by "Given the asymmetric dependency transgender people often feel from their doctors." Are the authors referring to social desirability bias? Please clarify.

OVERALL

Several times throughout the manuscript, statements were made in parenthesis. However, these statements were important for understanding the main point and the parenthesis were distracting. I recommend removing as many parenthetical statements as possible and incorporating them into the main text.

Reviewer #3:

Remarks to the Author:

This is a well-written manuscript that systematically reviewed differences in well-being, self-mastery, and interpersonal functioning following gender affirming hormone therapy (GAHT) among transgender people taking masculinizing and feminizing therapies. The manuscript included both qualitative (n=5) and quantitative (n=27) empirical studies. Of the quantitative studies, 12 were cross-sectional and 15 were prospective cohort studies. Studies from 1983 to the present were included; there were no geographic restrictions.

The breadth of time and location as well as the differential size, power, type, and rigor of the included studies that are compared and contrasted in the review undermine the impact of the findings. For example, results from 1 small poorly designed and underpowered study from 1 country are compared to results from three other studies that are larger and with stronger designs from other countries. The result is that it is difficult to interpret the reliability of meaning within the review. The abstract in particular is an example of this: Findings across the review are summarized in the abstract without noting differences in rigor that are described elsewhere in the manuscript.

The last paragraph of the results and the discussion is the strongest part of the manuscript. In particular, the last paragraph of the results acknowledges that it is not possible to tell from the extant

literature what factors (biological, psychological, or sociocultural) therapy might influence psychosocial outcomes related to hormone use. The discussion also notes the differences and limitations in rigor of findings. These limitations should be clarified in the abstract as well.

The worst part of the manuscript is the introduction. The introduction is stigmatizing in that it appears to imply that poor psychosocial outcomes among transgender people (lines 62-72) could in part be due to side effects of GAHT (see lines 98-109). The authors state that studies investigating the “biological effects” of GAHT on psychosocial function of transgender people should be a top research priority. They state this despite the fact that the strongest findings in their own review show that GAHT is associated with improved wellbeing.

The discussion notes that studies on transgender health should be undertaken with transgender communities. Have the authors done any engagement with transgender communities to inform this study? If not, I recommend that the authors do so. I strongly suspect this authorship team will soon find that a focus on assessing the “biological effects of hormone therapy on psychosocial outcomes” to warn transgender people of possible negative side effects of GAHT is not in fact a top research priority of transgender communities. In fact, the entire research focus described in the introduction may be seen as offensive by many transgender communities, especially given that GAHT is often lifesaving. As the authors again describe in the discussion section of their own manuscript, teasing apart the biological from the psychological and cultural factors that shape psychosocial functioning of transgender people receiving GAHT is inherently confounded. The authors should seek feedback from transgender community stakeholders and revise this manuscript extensively in response.

Author Rebuttal to Initial comments

Reviewer 1

First, there have been additional studies that have been published on this topic in the last year and the authors should update accordingly. Also, there were articles that were missed and the authors should do additional searches to ensure they have covered more of the literature.

Our Revisions:

As noted in our response to the editor, the lack of update primarily had to do with the time the paper was initially submitted. We have now updated our search and included papers published or in press by the date we received the letter requesting revisions (i.e., May 2022).

Reviewer 1

Finally, the authors need to be mindful of making conclusions or sweeping comments about findings without having meta-analytic data to back up the assertions.

Our Revisions:

As in our response to the editor above, we have gone back through the manuscript and attempted to identify and revise any claims that we believed needed greater evidence for support. We have also revised any sentences specifically identified by the reviewers in this fashion.

Reviewer 1

In general, instead of using binary language of transmasculine and transfeminine, I recommend focusing on the

naming the type of hormones that people would receive, for example: masculinizing hormones or feminizing hormones, given the number of nonbinary, agender, genderfluid, etc. people who will likely be included in your review.

Our Revisions:

We agree with the reviewer's preference to avoid binary language. We had generally structured the review already as suggested according to masculinizing and feminizing hormone therapy outcomes and we have now gone through the manuscript to revise using this language instead of transmasculine and transfeminine where relevant (primarily in the abstract and the results of qualitative studies as well as the discussion).

Reviewer 1

I recommend including nonbinary populations in the abstract and in the review, given the number of nonbinary people who access gender affirming hormones. See line 16 where only transgender people are mentioned.

Our Revisions:

Thank you for this suggestion. We agree that our review includes those who identify as non-binary and gender diverse (as can be seen in study descriptions in Table 1). We have revised the abstract to clearly state that we use transgender here as an umbrella term that includes those who identify as non-binary, genderqueer etc., which is also stated in the introduction when defining transgender as well as referenced in our search terms and elsewhere in the manuscript.

Reviewer 1:

The authors mention 32 papers—I'm curious as a reader at this point if this means journal articles, book chapters, and nonpublished papers (dissertations) or if the authors mean journal articles only. See line 16.

Our Revisions:

Thank you for pointing out our lack of clarity here. We have revised this statement to read "journal articles." Please see our response to the editor for further details about why we restricted our search to the published literature.

Reviewer 1:

On line 20, what does "more mixed" mean—more than what?

Our Revisions:

We have revised this statement from "more mixed" to "somewhat inconsistent" in order to better capture our meaning and remove the need for a comparator.

Reviewer 1:

The commentary on why studying psychosocial functioning as an outcome of hormone therapy is compelling and demonstrates that importance of this study. As a minor note, the word "but" on line 44 almost downplays your argument—I would just simply state the additional outcome of psychosocial states and social interactions.

Our Revisions:

Thank you for this comment, we agree the argument is compelling and have removed the word "but" as suggested to avoid downplaying this.

Reviewer 1:

Are the studies starting on line 60 only focused on cisgender populations?

Our Revisions:

We have revised the text to specify that these studies were conducted with cisgender participants specifically.

Reviewer 1:

I am always a bit concerned in literature reviews when disparities are discussed without the theoretical explanation for why they exist in the first place (e.g., minority stress causing significant levels of social anxiety). I recommend at least contextualizing the disparities that are discussed starting on line 62.

Our Revisions:

We strongly agree that social stigma and minority stress for transgender people are largely responsible for shaping disparities in social relationships. Our past work has focused on this exact question and we have now revised the text to specify that these outcomes are “likely driven in part by experiences of stigma and invalidation.”

Reviewer 1:

I'm curious about how relevant the Bos et al. 2010 study is, given that cis women were only given sublingual testosterone on two separate days and then given facial trustworthiness evaluations—I would recommend that citations that follow people who take hormones for a period of time be included as evidence in this study, given the arguments you make throughout your manuscript.

Our Revisions:

While we do think that research on acute effects is relevant to making inferences about possible long-term effects of exogenous hormone administration, we agree with the reviewer that this is probably less relevant than evidence for long-term administration. Therefore, we have revised the manuscript by removing reference to the Bos et al. (2010) study. Other references that we have included in this section more specifically follow hormone administration over a period of time. We also now include reference to the effects of puberty in the introduction, which also speaks to this point.

Reviewer 1:

I'm wondering where the citation is for this assertion: “Currently, transgender people are routinely prescribed such biological interventions without being made sufficiently aware of potential psychosocial implications (and available counselling options to support positive adjustment), potentially risking social disruption and, therefore, poorer health and well-being across the life course.” See line 98.

Our Revisions:

Given the concerns of the reviewers regarding this particular sentence, we have now carefully revised it to read, “Clinical guidance for the provision of gender-affirming care, particularly gender-affirming hormone therapy, relies on a relatively underdeveloped and variable evidence base with little acknowledgement of potential psychosocial implications (Dahlen et al., 2021; Salas-Humara et al., 2019).” We feel that this better reflects our meaning and is based on evidence from two reviews of various clinical guidance papers from key professional groups around the world.

Reviewer 1:

It would be helpful to specify the aims further—you note on line 114: “That is, we aimed to understand not only what effects hormone therapy might have on psychosocial functioning, but also whether these could be unambiguously and directly related to the hormonal changes, or whether existing evidence does not separate this from the various other changes that are associated with gender identity transition.” It might be useful to mention how you are able to determine the second part of this aim or specify further.

Our Revisions:

Given that our secondary aim was to attempt to gauge to what extent the current literature on this topic could be used to identify direct causal associations, we have added a further sentence specifying that, “To do so, we also formally assessed risk of bias for each study included in our review as well as coding for potential confounding variables in order to gauge which were most often included in the quantitative literature on this topic.” This revision helps clarify that our review both assesses the current literature and describes potential bias and confounding in further detail.

Reviewer 1:

The data for this study are a year old—most journals will not publish meta analysis or systematic reviews unless the data are analyzed within 3 months of submissions. I recommend this gold standard for this particular review as well.

Our Revisions:

Please see our earlier response to the editor on this point. In short, we have updated our search and included all articles published or in press by the date of receipt of the revision letter (i.e., May 2022).

Reviewer 1:

What does this mean on line 126: “encompass normal levels of functioning/symptomatology and not focus exclusively on clinical diagnoses/thresholds”

Our Revisions:

This inclusion criterion simply means that we did not include studies that only assessed changes in clinical psychiatric diagnoses. We have revised the manuscript to more clearly state, “encompass normal levels of functioning—that is, we chose not to include studies that examined changes in clinical psychiatric diagnoses.”

Reviewer 1:

Were only journal articles used in this review? And if so, why? And what does that mean for the file drawer effect?

Our Revisions:

We did indeed restrict our search to the published academic literature, as we have revised the search strategy section to describe more explicitly. We also outline our two primary reasons for this choice, which we have described above in our response to the editor.

Reviewer 1:

It seems like being more specific in the qualitative section would be beneficial to this review. Again, I suggest having two sections—one regarding feminizing hormones and one regarding masculinizing hormones. Given the second aim of the study, I’m wondering if the authors looked into determining how findings could delve into the complexity of outcomes related to hormone therapy. Also, I wonder if emailing the authors for the Riggs study to determine if there were quotes related to individuals taking masculinizing hormones are available for the review?

Our Revisions

The use of “transfeminine” and “transmasculine” headings in this section may have led to some confusion as we had previously separated these sections as recommended by the reviewer by those on feminizing and masculinizing hormones (and indeed, at least one study included a non-binary person here). As described previously, we have revised these headings and the text to use the terms “those on feminizing and masculinizing hormone therapy.” Related to the findings, we have attempted to draw out the complexity as much as possible, but ultimately this is limited by the number of studies on this topic as well as the fact that almost all of these studies only included one theme devoted to this topic rather than a whole paper on psychosocial effects of hormones. We concluded that expanding beyond what is currently available, or suggesting that results were more conclusive in this way, would be somewhat misleading. The brevity of this section to some extent reflects the paucity of this work. However, we have

been able to add one additional study (Ussher et al., 2022) here after our updated search. We have also revised the reference to the Riggs study as recommended by the reviewer to more clearly reflect that although none of the quotes were from those on masculinizing hormones, the experiences reported on in the theme were suggested to refer to transgender participants in the study as a whole rather than only those on feminizing hormone therapy as reported by the authors.

Reviewer 1:

In reading the cross-sectional section, I'm wondering if there was a cap at the timeframe for when studies were published? It seems that the process of administering hormone therapy has changed over the last several decades—it might be worthwhile to note or comment on some of the older studies vs the new ones—the Blanchard 1983 study stands out compared to the other relatively newer ones.

Our Revisions

As described in the method section, we included any studies published between 1980 and May 2022. This choice was informed in part by findings of past reviews (e.g., Nobili et al., 2018) that did not locate any papers further than 40 years back even with wider search dates as well as an attempt to incorporate as much published research as possible in our review. We agree that it is important to acknowledge how the quality of work has shifted over time, and in our revision we are now able to do this given the coding of risk-of-bias for all studies. As our revisions throughout the results highlight, indeed some of the older studies (including Blanchard et al., 1983) were of lower quality.

Reviewer 1:

But also, as I read through the review, I noticed some studies were missed, for example this one and subsequent ones out of this dataset:

Keo-Meier, C. L., Herman, L. I., Reisner, S. L., Pardo, S. T., Sharp, C., & Babcock, J. C. (2015). Testosterone treatment and MMPI-2 improvement in transgender men: A prospective controlled study. Journal of consulting and clinical psychology, 83(1), 143.

Our Revisions:

We have revised our search strategy and delved further into specific references from related reviews to attempt to capture any relevant literature that we may have missed. In addition to the papers identified from the interim period since our manuscript was under review we have identified 7 further papers that were not included in the prior version of the manuscript. We have compiled all of these new papers into the revision and synthesized their findings in the revised results.

With regards to the specific reference listed here, we did not include this paper as it did not meet our criteria for inclusion in the current review. Specifically, we did not include any papers using the MMPI as this clinical instrument involves a potentially problematic sex-standardization process given that the sex norms have been validated with exclusively cisgender samples (Webb et al., 2016) as we now describe in our revised inclusion criteria. This was also true of other papers from this group/dataset. However, in responding to this comment we also noticed that our reference manager transposed the surname of the first author of this paper to “Colton Meier” (whereas the name is actually Colton Keo-Meier). Therefore, two other relevant papers that include this author were already in our review, but they may have been overlooked due to this error. We have revised the manuscript now to correct this author’s name throughout.

Reviewer 1:

This seems like a leap for a systematic review, see line 417: “Interestingly, this happens irrespective of whether the treatment is feminizing or masculinizing, which might be taken to suggest that these well-being improvements are

not necessarily directly due to hormonal changes, but to the positive effects of transition on body image (or other confounders), which in turn improves well-being (or even simply a reflection of the positive sense that having control and autonomy over one's body and gender expression brings)." Unless the authors are willing to analyze the data in aggregate, it seems impossible to make this assertion from systematic review. There are confounding variables for every medical (and psychological) study and this assertion could be made for pretty much anything—without actually having data, it seems impossible to know what the direct effects of treatments might be.

Our Revisions:

We agree that the wording of this sentence did not convey our meaning very well and have revised it given the greater nuance of our findings in the updated review. We now focus on the fact that these improvements in well-being seem to be driven by reductions in distress: "Notably, these changes all reflect reductions in distress rather than direct increases in positive states, suggesting that gender-affirming hormones may improve well-being primarily by helping to eliminate gender dysphoria, whether through improved body image or other relevant pathways, or even simply increasing sense-of-control and autonomy over one's body and gender expression." This conveys the limitation of confounding inherent in the reviewed studies but eliminates the speculation regarding similarities between masculinizing and feminizing hormone therapies, which as the reviewer highlighted is not examinable in the current review.

Reviewer 1:

I would suggest that the authors do a deeper dive into the section starting on page 422. It might be useful to pose additional questions about blunted affect as a result of testosterone—does this cause distress to the people experiencing it? Similar with mood swings and emotional imbalances—it seems like posing research questions around these would be useful—are these clinical levels emotional changes or are they changes in people actually experiencing their emotions more deeply for the first time or in a way they haven't in a while?

Our Revisions:

We agree that this is an interesting finding and have revised this paragraph in the discussion to include further consideration of it as suggested by the reviewer. Specifically, we now relate our findings to further results from ENIGI that were not eligible for inclusion in our review but point in the same direction. As the reviewer suggests, we then pose questions for future research, including the duration of any such changes and whether they are experienced as distressing by transgender people.

Reviewer 1:

On line 445, it would be helpful to mention specific measures that researchers could use.

Our Revisions:

We appreciate this suggestion and its utility for future work, therefore we now point toward the UCLA Loneliness Scale, Rempel and Holmes Trust Scale, Experiences in Close Relationships Scale and Couples Satisfaction Index as potential measures to include in order to tap into the interpersonal functioning dimension of psychosocial functioning in more diverse ways.

Reviewer 2:

The phrase, "...with hints toward positive change.." is colloquial. Scientific language such as "trend toward positive change" is more appropriate.

Our Revision:

We have revised this sentence in the abstract in line with the reviewer's suggestion.

Reviewer 2:

The first paragraph is missing several citations and some of the citations provided are quite dated. For example, one reference referred to data from 2010-2014. Citations are needed to support the following statements: (1) "most transgender people continue to use some dosage of gender-affirming hormones throughout their lives" (2) "hormones also affect psychological states and social interactions" (3) "large and growing prevalence of gender-affirming hormone therapy across countries"

Our Revisions:

Citations for each of these three statements have now been added in the revised manuscript. Regarding the dates of the specific study mentioned by the reviewer, this study by Wiepjes and colleagues was conducted and published in 2018. The dates of 2010-2014 mentioned in the text are when people were first referred to the clinic, but the study follows whether they began hormone therapy over the subsequent five years (i.e., 2014-2018), which is more relatively recent and the highest quality study available for this point.

Reviewer 2:

The statement at the end of the first paragraph that gaps in knowledge limit fully informed consent is inaccurate. Informed consent processes require sharing with the client what is known and unknown about a procedure or treatment. It does not require that everything be known in order to be fully informed consent.

Our Revisions:

We agree that our point regarding informed consent was not fully articulated in this brief clause and have therefore removed it in line with the reviewer's suggestion.

Reviewer 2:

The Testa 2017 citation at the end of the second paragraph does not support the claim that suicide risk in trans people is linked to disruption in their social lives. The cited manuscript links minority stressors such as rejection, non-affirmation, victimization, and discrimination, to suicide risk. If this is what is meant by "disruption in social lives" please be specific about that in the manuscript.

Our Revisions:

It is correct that the Testa et al. 2017 paper looks at rejection, non-affirmation, victimization and discrimination (i.e., minority stressors), but critically, it also tests "interpersonal theory factors" in the form of thwarted belongingness and perceived burdensomeness, which are conceptually similar to what we have framed as disruption in social lives—therefore, we have retained this citation in support of the point here. To further support this point, we have added an additional citation (Branstrom et al., 2022) to another recent study that demonstrates links to social-relational factors, including poor social integration and interpersonal risks, as predictors of suicidality in transgender people.

Reviewer 2:

The third paragraph provides definitions of terms that have already been used throughout the first 2 paragraphs. I recommend moving it up to be the first paragraph.

Our Revisions:

We have revised this paragraph by moving the definitional element (i.e., our use of transgender as an umbrella term) to the first paragraph as suggested by the reviewer.

Reviewer 2:

The 5th paragraph makes a strong claim that essentially insults all prescribers of gender affirming hormones: "Currently, transgender people are routinely prescribed such biological interventions without being made sufficiently aware of potential psychosocial implications (and available counselling options to support positive adjustment),"

potentially risking social disruption and, therefore, poorer health and well-being across the life course." Please provide a citation supporting this claim that prescribers do not make transgender people aware of known psychosocial implications. The fact that this is not described in current guidelines does not mean that it does not routinely happen.

Our Revisions:

As described in our response to Reviewer 1, we have edited this sentence to better reflect the limitations of current guidelines rather than specific behaviors of providers (which we agree could be read as insulting).

Reviewer 2:

In the final sentence of the Introduction, I recommend removing "identity" from the sentence. Trans people on hormone therapy undergo gender transition (of their bodies) but not a transition of their identity.

Our Revisions:

We have revised this sentence as recommended by dropping the word "identity." Furthermore, we have attempted to change any other references to "gender identity transition" to "gender transition" throughout the manuscript.

Reviewer 2:

Section 2.2: Who is "they" in the first sentence?

Our Revisions:

This was a typo. We have revised the sentence to read, "when carrying out the search strategy for this review."

Reviewer 2:

Please provide an exact list of the terms used for the search, including MESH terms, rather than supplying only examples. A table of search terms may be best for this.

Our Revisions:

To clarify this point for readers, we have now included a table in the appendix (Appendix 2) that extensively describes our search strategy, including exact terms and the versions of these that were used at each step in the process.

Reviewer 2:

Instead of general statements such as "large" and "small" sample sizes for the quantitative studies, please provide the exact sample sizes of the studies in the text when discussing their findings. Please note if any of the studies provided a power analysis, particularly those with null findings that may have been related to inadequate power to detect a significant difference.

Our Revisions:

We appreciate the reviewer's request for greater specificity in terms of sample size and we have revised the results accordingly, pointing to actual numbers rather than using terms such as small and large. Furthermore, we have included a column specifying whether each study included a formal power analysis in our assessment of risk of bias in the appendix. We have further revised the results to include mention of whether a given study included a power analysis where most relevant as requested by the reviewer.

Reviewer 2:

In section 3.2.1.1.1, the penultimate sentence states that comparisons between studies were "unclear" due to use of different measures. However, it seems more appropriate to state that comparisons across studies with different

measures is not valid.

Our Revisions:

We agree with the reviewer that the term “unclear” may not have been ideal here, but we would not go so far as to say that the difference in measures makes it such that any comparisons are “not valid.” Therefore, we have chosen to revise this sentence by simply removing this clause and allowing the reader to interpret the fact that the studies used different measures as they prefer.

Reviewer 2:

I recommend replacing potentially offensive term, "black market" with a term like "informal sector" or "from unlicensed providers."

Our Revisions:

We appreciate this suggestion and have revised to read “unlicensed provider” instead of “black market.” We have changed this elsewhere in the manuscript as well.

Reviewer 2:

The first sentence refers to "state of the art"; however "state of the science" is more appropriate in this context.

Our Revisions:

We have revised this sentence as suggested by the reviewer to read “state of the science.”

Reviewer 2:

The first sentence of the 3rd paragraph refers to "ambiguous results"; however, "inconsistent" or "inconclusive" would be a more appropriate term.

Our Revisions:

We have revised this sentence using “inconclusive” in place of “ambiguous.”

Reviewer 2:

Citations are missing for multiple statements of fact in the third paragraph. The second sentence refers to multiple earlier studies but does not cite any studies at all. The next sentence refers to confirmation in a more recent study but does not cite any study. It's not clear if this sentence: "However, anger intensity does not seem to increase for those on either masculinizing or feminizing hormone therapy." refers to findings from this review or to earlier studies; if it refers to earlier studies, a citation is needed. In the following paragraph, no citation is provided for this statement: "Evidence for improvements in this domain is limited and inconclusive, but hints toward positive change, once again for those on both feminizing and masculinizing hormone therapy (most commonly in the form of reduced social anxiety)."

Our Revisions:

These statements are all based on our summaries of the evidence present in the current review, not from specific studies. We have revised the discussion section extensively and attempted to make this clearer for the reader. For any findings that do not relate to the results of the current review, we have provided a relevant citation.

Reviewer 2:

In the 5th paragraph, is the author indicating that self-report bias is based on internalized stereotypes? If so, please state this clearly. If not, it's unclear how self-report bias is related to the discussion of stereotypes earlier in the paragraph.

Our Revisions:

We agree with the reviewer that this idea was not clearly stated, so we have revised the sentence to read, "If these changes, identified on self-report measures, do actually reflect gender-stereotyped expectations on the part of participants rather than objective changes in psychosocial functioning, other types of tasks that do not rely on self-report exclusively (e.g., behavioral tasks) may prove useful in unpacking these preliminary findings."

Reviewer 2:

In the 7th paragraph, please indicate the source for this quote, "are where they want to be." If this is not a quote, then I recommend replacing this colloquialism with the statement that is in parentheses after the quotation marks.

Our Revisions:

We have revised the paper to remove this colloquialism (i.e., "are where they want to be") as suggested by the reviewer.

Reviewer 2:

The authors appropriately note that waiting for hormone therapy may cause distress that could serve as a confounder when comparing those waiting with those currently on gender affirming hormone therapy. They fail to note that there may ALSO be significant psychosocial differences (confounders) between trans people who are on hormones and those who do not want hormones, making that comparison group also potentially problematic.

Our Revisions:

We agree with this idea and had previously described the difficulty identifying an appropriate control group in such studies. To further establish this idea, we have revised this section to state, "However, it is of course possible that there are already differences in psychosocial functioning between people who do and do not seek hormone therapy to affirm their genders, meaning that studies utilizing both types of control groups and following changes in each over time may be most appropriate." We believe this statement more explicitly captures the meaning of our recommendation.

Reviewer 2:

In the 8th paragraph, I recommend replacing "in their full glory" with more professional language.

Our Revisions:

In an attempt to utilize more professional language as recommended by the reviewer, we have revised this to read "... which subsequently subsided to reveal its positive effects over time."

Reviewer 2:

In the 10th paragraph, it's not clear what is meant by "Given the asymmetric dependency transgender people often feel from their doctors." Are the authors referring to social desirability bias? Please clarify.

Our Revisions:

We acknowledge that this wording was somewhat unclear and have revised the sentence to read, "Given the authority of medical providers to determine treatment courses, some transgender people might wish to report effects that are in line with what is expected by providers so as to ensure the continuation of treatment."

Reviewer 2:

Several times throughout the manuscript, statements were made in parenthesis. However, these statements were important for understanding the main point and the parenthesis were distracting. I recommend removing as many parenthetical statements as possible and incorporating them into the main text.

Our Revisions:

We have considered this recommendation and removed parenthetical statements, incorporating them into the main text, whenever possible throughout the manuscript.

Reviewer 3:

The breadth of time and location as well as the differential size, power, type, and rigor of the included studies that are compared and contrasted in the review undermine the impact of the findings. For example, results from 1 small poorly designed and underpowered study from 1 country are compared to results from three other studies that are larger and with stronger designs from other countries. The result is that it is difficult to interpret the reliability of meaning within the review. The abstract in particular is an example of this: Findings across the review are summarized in the abstract without noting differences in rigor that are described elsewhere in the manuscript. The last paragraph of the results and the discussion is the strongest part of the manuscript. In particular, the last paragraph of the results acknowledges that it is not possible to tell from the extant literature what factors (biological, psychological, or sociocultural) therapy might influence psychosocial outcomes related to hormone use. The discussion also notes the differences and limitations in rigor of findings. These limitations should be clarified in the abstract as well.

Our Revisions:

In terms of the variability of study quality and rigor, we agree with the reviewer that this is a limitation of the literature. We have attempted to systematize our consideration of relevant factors in our revision by including a formal assessment of risk of bias as well as reference to exact sample sizes and whether or not studies included a formal power analysis, all of which we describe in detail in our response to other editor/reviewer comments. These revisions collectively make it somewhat easier for the reader to understand similarities and differences in findings and therefore increase the impact of our review. As the reviewer suggested, we have now revised the abstract to more clearly address the limitations regarding variability of study quality.

Reviewer 3:

The worst part of the manuscript is the introduction. The introduction is stigmatizing in that it appears to imply that poor psychosocial outcomes among transgender people (lines 62-72) could in part be due to side effects of GAHT (see lines 98-109). The authors state that studies investigating the “biological effects” of GAHT on psychosocial function of transgender people should be a top research priority. They state this despite the fact that the strongest findings in their own review show that GAHT is associated with improved wellbeing.

Our Revisions:

We appreciate the reviewer’s concerns here and have revised the manuscript (particularly the introduction) to avoid any potentially stigmatizing language or arguments. We also now include reference to minority stress as an important determinant of psychosocial functioning for transgender people, as described in our response to the previous reviewer.

Furthermore, we reiterate that the idea for this work was indeed developed collaboratively with transgender people and community stakeholders. Specifically, the lead author previously led a seed grant titled, *Transforming knowledge about the outcomes of gender identity service use: Co-development of a research agenda*, in which some of these ideas were initially discussed in focus groups and interviews with transgender people and members of their social networks. Furthermore, after this inspiration, an early draft of this paper was shared with a transgender (non-binary) colleague for comments and feedback. These were all incorporated into the first draft of the manuscript.

We do also think it is important to understand any potential biological effects of gender-affirming hormone therapy and plenty of previous theory/data support the notion that exogenous administration of hormones can shape psychosocial functioning (at least in some specific ways), with the results of our review potentially supporting this

idea. While the reviewer is correct that our review found improvements in well-being after gender-affirming hormone therapy, as we now more precisely describe in the revised manuscript, this was generally via reductions in distress rather than increases in positive states (e.g., increased quality of life). Other changes were much less clear-cut (e.g., potential reductions in social anxiety but accompanying increases in aggressive responding for those on masculinizing hormone therapy as well as increased emotional instability for those on feminizing hormone therapy but blunted affect for those on masculinizing hormone therapy). Consequently, we have revised the conclusion to more specifically flag the relevance of investigating the subcomponents of self-mastery and interpersonal functioning. As another reviewer points out, these findings are important and require further investigation, and we hope that our review will prompt that work to clarify some of these effects, whether they are primarily driven by biological changes or gender stereotypes, as we discuss in our revised manuscript. Importantly, we do not assign valence to most the psychosocial effects we examine—e.g., anger as possibly experienced as positive (i.e., as assertiveness/self-confidence).

Reviewer 3:

The discussion notes that studies on transgender health should be undertaken with transgender communities. Have the authors done any engagement with transgender communities to inform this study? If not, I recommend that the authors do so. I strongly suspect this authorship team will soon find that a focus on assessing the “biological effects of hormone therapy on psychosocial outcomes” to warn transgender people of possible negative side effects of GAHT is not in fact a top research priority of transgender communities. In fact, the entire research focus described in the introduction may be seen as offensive by many transgender communities, especially given that GAHT is often lifesaving. As the authors again describe in the discussion section of their own manuscript, teasing apart the biological from the psychological and cultural factors that shape psychosocial functioning of transgender people receiving GAHT is inherently confounded. The authors should seek feedback from transgender community stakeholders and revise this manuscript extensively in response.

Our Revisions:

Please see our response to the editor. In short, we have indeed engaged with transgender communities in conducting this work. In addition to past engagement, we have further engaged with community members/stakeholders for comments and insights on the most recent draft of the manuscript. Specifically, we shared the current draft with a transgender person who was both a participant in our formative work under the seed grant and is involved in running a charity providing care for transgender people in the UK as well as with a local gender identity service user who is transgender and volunteers with another key charity for the LGBTQ+ community in our region. These two community members have different experiences in terms of masculinizing versus feminizing hormone therapy and in terms of generational status. Their feedback has been incorporated into the revision with great care.

While we agree with the reviewer that gender-affirming hormone therapy can be viewed as “lifesaving” medicine, we do not think that this precludes careful investigation of potential outcomes/side-effects (particularly psychosocial). Our aim is to improve knowledge and practice in gender-affirming medicine and we have done our best to make this clear in our revised manuscript. Of particular note, as described in our earlier responses, we have revised the manuscript based on feedback from transgender community members in order to push back against the idea that the current state of the literature should be used as a justification for restricting access to gender-affirming hormone therapy. We see our review as a contribution to the argument that the reviewer points out—gender-affirming hormone therapy improves well-being (primarily reducing distress), and any other potential effects are certainly less clear-cut and do not off-set these important benefits of treatment.

Decision Letter, first revision:

23rd January 2023

Dear Dr. Doyle,

Thank you for submitting your revised manuscript "A Systematic Review of Psychosocial Functioning Changes after Gender-Affirming Hormone Therapy among Transgender People" (NATHUMBEHAV-22020422A). It has now been seen by the original Reviewer 2 and a newly recruited reviewer (Reviewer 4), and their comments are below. As you can see, the reviewers find that the paper has improved in revision. We will therefore be happy in principle to publish it in Nature Human Behaviour, pending minor revisions to satisfy the referees' final requests and to comply with our editorial and formatting guidelines.

We are now performing detailed checks on your paper and also consulting with a relevant advocacy group, who we are asking for input on the language and implications of this work for the community. We will send you this feedback, as well as a checklist detailing our editorial and formatting requirements within three weeks. Please do not upload the final materials and make any revisions until you receive this additional information from us.

Sincerely,

Samantha Antusch

Samantha Antusch, PhD
Senior Editor
Nature Human Behaviour

Reviewer #2 (Remarks to the Author):

Thank you for the opportunity to review this revised manuscript. I appreciate the authors' careful attention to reviewer comments and the revised manuscript it much improved. This review addresses an important concern in transgender health and will make a contribution to the literature on the effects of gender affirming hormone therapy.

Reviewer #4 (Remarks to the Author):

A review of studies of the effect of hormone treatment on transgender individuals psychosocial functioning is needed. This is an important field of research in need of a thorough review. The study is well conducted and described. In particular, I appreciate the thorough assessment of the included studies and the attempt to assess risk of bias of the included study. I have some minor comments that I believe would improve the quality of the manuscript.

Introduction

The authors claim that "... decades of research in humans and non-human animals have confirmed that hormones strongly influence psychosocial functioning via biological pathways" but provide references to studies that don't provide a strong support for this claim. Rather, the field is mixed regarding the effect of hormone on psychosocial functioning and the authors should acknowledge the limited knowledge in the field and the need to better understand these patterns and their influences and consequences.

Similarly, the author claims that: "... exogenous administration of testosterone affects many aspects of psychosocial functioning in cisgender men and women, including increasing social aggression" but refer to a study that concludes that: "... results are conflicting and inconclusive." The authors need to be more accurate in their reporting of what previous studies have found and not found.

Method

The identification of relevant studies seems to have been adequately performed and is well described.

Results and discussion

The results are well described and accounted for and the discussion and conclusion adequate. In conclusion, the manuscript makes an important contribution to knowledge in the field.

Final Decision Letter:

Dear Dr Doyle,

We are pleased to inform you that your Article "A Systematic Review of Psychosocial Functioning Changes after Gender-Affirming Hormone Therapy among Transgender People", has now been accepted for publication in *Nature Human Behaviour*.

Please note that *Nature Human Behaviour* is a Transformative Journal (TJ). Authors whose manuscript was submitted on or after January 1st, 2021, may publish their research with us through the traditional subscription access route or make their paper immediately open access through payment of an article-processing charge (APC). Authors will not be required to make a final decision about access to their article until it has been accepted. IMPORTANT NOTE: Articles submitted before January 1st, 2021, are not eligible for Open Access publication. Find out more about Transformative Journals

Authors may need to take specific actions to achieve compliance with funder and institutional open access mandates. If your research is supported by a funder that requires immediate open access (e.g. according to Plan S principles) then you should select the gold OA route, and we will direct you to the compliant route where possible. For authors selecting the subscription publication route, the journal's standard licensing terms will need to be accepted, including self-archiving policies. Those licensing terms

will supersede any other terms that the author or any third party may assert apply to any version of the manuscript.

With best regards,

Samantha Antusch

Samantha Antusch, PhD
Senior Editor
Nature Human Behaviour